# Special Education Teachers: The Role of Autonomous Motivation in the Relationship between Teachers’ Efficacy for Inclusive Practice and Teaching Styles

**DOI:** 10.3390/ijerph191710921

**Published:** 2022-09-01

**Authors:** Alessia Passanisi, Caterina Buzzai, Ugo Pace

**Affiliations:** 1Faculty of Human and Social Sciences, UKE-Kore University of Enna, Cittadella Universitaria, 94100 Enna, Italy; 2Faculty of Classic, Linguistic and Educational Studies, UKE-Kore University of Enna, Cittadella Universitaria, 94100 Enna, Italy

**Keywords:** self-efficacy, teaching, inclusive practices

## Abstract

Research on special education teachers has increased in recent years. However, few studies have investigated factors related to teachers’ preference for a specific style in inclusive education. For this reason, the aim of this cross-sectional study is to investigate the mediating role of autonomous motivation in the relationship between teachers’ efficacy for inclusive practice (TEIP) and teaching styles (structuring, autonomous, controlling, chaotic). Four hundred and twenty-three pre-service special education teachers participated in the study. Participants were administered the following self-reports: TEIP scale, Autonomous Motivations for Teaching Scale, and the Situations-in-School questionnaire. SEM analysis confirmed the role of autonomous motivation as a mediator for TEIP on teaching styles. Moreover, the results showed a positive association between TEIP and both autonomy and structuring teaching styles. The findings of this study suggest the importance of implementing specific special-education teacher training to promote intrinsic motivation toward teaching in an inclusive context.

## 1. Introduction

Inclusive education requires schools to be capable of meeting the needs of any student, including those with special education needs, and supporting their learning and involvement in class activities. According to Italian legislation, students with special education needs (i.e., disabilities; specific developmental disorders; and socio-economic, linguistic or cultural disadvantage) must be encouraged to attend mainstream education at all levels.

Offering high-quality education for all students is a duty that depends on the quality of teacher education [1], as teachers are responsible for developing and implementing proper strategies for each student’s learning process.

Effecting inclusive education necessitates changes in perspective and the investigation of new pedagogical strategies so that everyone can contribute and learn together, increasing the quality of vertical and horizontal relationships in class.

The advantages of teacher education to inclusive practices are related to the strengthening of teachers’ beliefs in self-efficacy [2].

According to Social Cognitive Theory [3], self-efficacy refers to the individual’s belief about their capacity to organize and execute the actions necessary to produce certain outcomes. In inclusive education, it refers to the ability of teachers to select educational-didactic objectives and to use teaching methodologies based on the characteristics of the students, preventing harassing behavior, and collaborating with families and other professionals [4]. The higher the perceived self-efficacy, the greater the amount of effort and perseverance is spent on a given assignment, since individuals tend to avoid tasks for which they think they do not possess the necessary abilities, choosing to allocate their resources to those in which they believe they are competent to gain success [3,5]. As a result, people become intrinsically motivated by their work. Teachers require a higher degree of self-determination and motivation to deal with their challenging educational and teaching demands.

DeCharms [6] differentiated between two kinds of perceived sources for intentional action: extrinsic and intrinsic. The author emphasized that people have extrinsic motivation when they perceive the source of initiation and regulation of their goal-directed actions to be external to themselves; as for intrinsic motivation, the locus of initiation and regulation is perceived to be internal. Accordingly, Ryan and Deci’s Self-Determination Theory (SDT) [7] substituted the extrinsic/intrinsic dichotomy with a more differentiated continuum of autonomous-versus-controlled motivations. Research has suggested that autonomous motivations allow people to fulfil their authentic selves [6,7,8], whereas controlled motivations are considered to be sources of external or internal pressure [7,9]. Moreover, several studies have shown a significant association between self-efficacy and motivation. Although self-efficacy has often been considered as a mediating variable between motivation and behavior, recent cross-sectional and longitudinal studies have suggested that self-efficacy has an effect on motivation [10,11], and that the latter explains the relationship between self-efficacy and teacher behaviors [12]. In addition, the construct of self-efficacy is similar to the need for competence described by the SDT [13]. In fact, both SDT [7] and Social Cognitive Theory [3] suggest that humans are agents of their actions and desire to feel competent and effective. Furthermore, the SDT has shown that the satisfaction of the need for competence contributes to more autonomous motivation [14].

Teachers’ beliefs in their self-efficacy are also associated with teaching styles that tend to promote the autonomy of students, assigning to them a central role in the educational process and perceiving their difficulties as removable with extra care; they create an encouraging atmosphere for learning and organize their teaching activities, dealing with didactic problems in a structured way [5,15]. On the contrary, teachers with low perceived self-efficacy are resilient to innovations, more likely to limit students’ autonomy, keeping an authoritarian attitude and adopting punitive disciplinary strategies to avoid efforts to deal with stressful problems and to negotiate less with their students’ demands. They have a pessimistic opinion of the educability of students, often blamed when learning outcomes do not appear to be meet expectations [3,5].

Two motivating teaching styles (autonomy-supportive and structuring), and two de-motivating teaching styles (controlling and chaotic) have been examined in the literature [16,17]. They vary in the extent to which they meet or frustrate students’ three basic psychological needs for autonomy, competence, and relatedness as posited by SDT [7,16,18]. A massive amount of research has dealt with the positive outcomes of autonomy-supportive or structuring styles and the negative effects of controlling or chaotic styles [19,20,21,22,23], while few researchers have investigated the factors prior to the adoption of a particular teaching style [16,24,25,26]. Among them, Vermote et al. [26] suggested that autonomously motivated teachers organize teaching material taking into account the interests and values of the students; they are open to the students’ point of view (supportive autonomy approach), clearly communicate their expectations, and provide help and encouragement (structuring approach). Furthermore, the authors suggested that autonomous motivation can act as a buffer against the adoption of a chaotic approach, that is, an attitude of renunciation and abandonment toward students [26]. In addition, with regard to inclusive education, Buzzai et al. [24] recently suggested that teachers’ efficacy for inclusive practices was a direct positive predictor of motivating teaching styles in multicultural classrooms. However, more research is needed regarding the factors related to teachers’ predilection for a specific style in inclusive education.

Therefore, the aim of this cross-sectional study is to investigate the relationship between teachers’ efficacy for inclusive practices (TEIP), autonomous motivation, and (de)motivating teaching styles. In the same vein as previous studies [10,12,24,26], it is sensible to hypothesize that teachers’ efficacy for inclusive practices (TEIP) enhances the autonomous motivation for teaching (i.e., teachers’ thoughts and feelings regarding their own motivations for engaging in teaching), and that teachers’ autonomous motivation increases motivating teaching styles and serves as a buffer against the adoption of demotivating teaching styles. According to this mechanism, autonomously motivated teachers desire their students to act and learn from autonomous motivations as they believe that these kinds of motivations lead to a high quality of learning and to an increased interest in the subjects they teach and love. Thus, autonomously motivated teachers use their own motivational experiences as a basis for assuming that students will engage in learning in the most thoughtful way if they understand the value of the learned subject and find it interesting. As a result, those teachers then primarily use supportive strategies (autonomy-supportive and structuring) actions, such as explaining the importance of various subjects to students’ achievements and allowing them to choose learning activities they appreciate, rather than controlling and chaotic strategies (i.e., the use of intrusive strategies or abandoning approach). In particular, in the present study a model concerning the mediating role of autonomous motivation in the relationship between TEIP and teaching styles is hypothesized. Accordingly, the following hypotheses have been tested:

**H1.** 
*Teachers’ efficacy for inclusive practices is positively associated with autonomous motivation.*


**H2.** 
*Autonomous motivation is positively related with motivating teaching styles (autonomy-supportive and structuring) and negatively related with demotivating teaching styles (controlling and chaotic).*


**H3.** 
*Teachers’ efficacy for inclusive practices is positively associated with autonomy-supportive and structuring teaching styles and negatively associated with controlling and chaotic teaching styles.*


**H4.** 
*Autonomous motivation mediates the relationship between teachers’ efficacy for inclusive practices and (de)motivating teaching styles.*


## 2. Method

### 2.1. Participants

Four hundred and twenty-three pre-service special education teachers participated in the study—92 males (22%) and 331 females (78%), with an average age of 38.06 years (SD = 7.08). We selected participants from specialization courses with support activities for secondary school at the university where they were studying in Enna, Sicily (Italy). Furthermore, all teachers had experience teaching students with special educational needs: most of them already worked as special education teachers; alternatively, they were all completing internships as special education teachers at school. In addition, all participants taught high school students both with and without special education needs.

### 2.2. Measures

A demographic questionnaire was used to collect the participants’ basic demographic information, including their age and gender and internship experiences.

The Teacher Efficacy for Inclusive Practice (TEIP) scale [4] was used to assess teacher self-efficacy in the context of inclusion. The TEIP was chosen because compared to other scales that assess the teacher’s self-efficacy such as the Teachers’ Sense of Efficacy Scale [27], it was specifically built to assess the ability of teachers to implement inclusive strategies. The scale consists of 18 items divided into three subscales: efficacy in managing behavior (e.g., “I am able to get children to follow classroom rules”), efficacy in collaboration (e.g., “I can collaborate with other teachers”), and efficacy in using inclusive instruction (e.g., “I am able to provide an alternate explanation or example when students are confused”). Items are rated on a four-point Likert scale, ranging from 1 (strongly disagree) to 6 (strongly agree). This scale has already been used in an Italian sample in a previous study, showing good reliability [28]. In this study, a total score was calculated, and the Cronbach alpha was 94.

The Autonomous Motivations for Teaching Scale [29] was used to assess motivations for teaching. This scale was chosen because rather than evaluating a relative autonomy index makes it possible to measure the different types of motivation as postulated by the SDT [30]. The scale includes 16 items divided into four subscales to investigate different types of motivation: external, introjected, identified, and intrinsic motivation. In this study, we used the subscale intrinsic motivation as an indicator of the most autonomous form of motivation (e.g., “When I invest effort in my work as a teacher, I do so because I enjoy finding unique solutions for various students”). Items are rated on a five-point Likert scale ranging from 1 (totally disagree) to 6 (totally agree). This scale has already been used in an Italian sample in a previous study showing good reliability [31]. In this study, the Cronbach alpha was 0.82.

The Situations-in-School (SIS) questionnaire [16] was used to assess teaching style. It was chosen because it presents participants with authentic situations and describes the four (de)motivating styles postulated by the SDT. The questionnaire consists of 15 vignettes followed by four potential behaviors that represent the four (de)motivating styles: autonomy-supportive (e.g., “Invite students to suggest a set of guidelines that will help them to feel comfortable in class”), structuring (e.g., “Make an announcement about your expectations and standards for being a cooperative classmate”), controlling (e.g., “Post your rules. Tell students they have to follow all the rules”), and chaotic (e.g., “Don’t worry too much about the rules and regulations”). The respondent must indicate on a seven-point Likert scale the degree to which each of these four behaviors described their own style ranging from 1 (does not describe me at all) to 7 (describes me extremely well). This scale has already been used in an Italian sample in a previous study showing good reliability [21]. In this study, the Cronbach alpha was 0.91 for autonomy-supportive, 0.92 for structuring, 0.88 for controlling, and 0.87 for chaotic.

### 2.3. Procedure

The test battery was completed by the participants after providing their personal informed consent in accordance with the Declaration of Helsinki [32]. The battery compilation took between 30 and 40 min, and the privacy and anonymity of their responses were guaranteed. The questionnaires were administered through Google Forms (online software that creates and edits surveys and allows the information collected to be automatically entered in a spreadsheet).

### 2.4. Data Analysis

Descriptive statistics, Cronbach’s alpha, and correlations were conducted using IBM SPSS 26.0 [33], and the structural equation modeling (SEM)—with latent variables—was carried out using RStudio [34] with the lavaan package [35]. The SEM approach was used because this reduced the probability of type I errors and is demonstrated to be superior to traditional univariate and multivariate approaches [36]. Moreover, according to the suggestions of Wu and Jia [37], confidence intervals (CIs) of the direct and indirect effects with 5000 bootstrap replication samples were used, and a 95% bias-corrected CI was applied. Several indexes were considered to prove the goodness-of fit: the Chi-square (χ^2^) value, Chi-square ratio (χ^2^/gl), the comparative fit index (CFI), the root mean square error of approximation (RMSEA) with its 90% confidence interval (CI), and the standardized root mean square residual (SRMR). The cut-off for a good model fit is achieved when the χ^2^/gl are ≤5, CFI values are >0.90, and the RMSEA and SRMR are <0.08 [36]. TEIP and SIS and were represented by four parcels, consisting of randomly selected items, while autonomous motivation was represented by four items of the scale [38].

## 3. Results

### 3.1. Descriptive Statistics, Reliability, and Correlations

The means, standard deviations, skewness, kurtosis, Cronbach’s alpha values, and correlations for all the variables considered in this study were shown in Table 1. The descriptive analysis showed that all scales have adequate scores of symmetry and kurtosis (skewness from −2 to +2, and kurtosis from −7 to +7) [39]. The reliability of all measures was reported to range from 0.82 to 0.94. Correlational analysis indicated that teacher efficacy for inclusive practice and autonomous motivation were positively correlated with autonomy-supportive and structuring teaching styles, and negatively correlated with controlling and chaotic teaching styles.

### 3.2. SEM Analysis

To investigate the mediating role of autonomous motivation in the relationship between the of teacher efficacy for inclusive practice and teaching styles (autonomy-supportive, structuring, controlling, chaotic), SEM analyses were employed.

The model estimation indicated that the data fit the model well (Figure 1), χ^2^ (237) = 633.11, *p* < 0.001, the χ^2^/gl = 2.67, CFI = 0.94, RMSEA = 0.06 (90% CI = 0.06–0.07), SRMR = 0.06, showing a direct positive associations from teacher efficacy for inclusive practice to autonomous motivation (β = 0.44, *p* ≤ 0.001, CI = 0.26, 0.54), to autonomy-supportive (β = 0.15, *p* ≤ 0.05, CI = 0.03, 0.32), and to structuring (β = 0.18, *p* ≤ 0.01, CI = 0.06, 0.35). Furthermore, direct positive relations were showed from autonomous motivation to autonomy-supportive (β = 0.56, *p* ≤ 0.001, CI = 0.53, 0.91), and to structuring (β = 0.52, *p* ≤ 0.001, CI = 0.42, 0.80). Instead, direct negative relations were showed from autonomous motivation to controlling (β = −0.22, *p* ≤ 0.01, CI = −0.43, −0.09), and to chaotic (β = −0.25, *p* ≤ 0.01, CI = −0.34, −0.08).

Finally, regarding the indirect effects, the results showed that autonomous motivation positively mediates the relationship between teacher efficacy for inclusive practice and autonomy-supportive (β = 0.24, *p* ≤ 0.001, CI = 0.18, 0.41) and structuring (β = 0.23, *p* ≤ 0.001, CI = 0.15, 0.35); and negatively mediates the relationship between teacher efficacy for inclusive practice and controlling (β = −0.10, *p* ≤ 0.01, CI = −0.19, −0.03) and chaotic (β = −0.25, *p* ≤ 0.01, CI = −0.15, −0.03) (see Table 2).

## 4. Conclusions

In the present study, the potential predictive role of TEIP on the adoption of teaching styles was examined using a mediational approach in which autonomous motivation was investigated as mediator of the TEIP–teaching styles relation. The findings confirmed the first research hypothesis, suggesting that TEIP was positively associated with autonomous motivation. Consistent with previous studies [10,11,12,40,41], the results indicated that teachers who feel able to implement inclusive teaching strategies are more involved in the activities they carry out because they are intrinsically motivated. Furthermore, this finding is consistent with SDT which suggests that the desire to feel effective and competent contributes to the development of autonomous motivation [14].

The findings of this study also confirmed the second research hypothesis, showing that autonomous motivation has a predictive role in teaching styles in a different way: while autonomous motivation was positively associated with structuring and autonomous teaching styles, it was negatively related to teachers’ chaotic and controlling styles. This result is coherent with the findings of researchers who have reported a similar pattern with the explored variables [26,42,43,44,45]. A possible explanation for this relationship is that autonomous motivation reflects the degree to which a teacher puts effort into and focuses on actual teaching, using autonomy-supportive and structuring styles [26]. In fact, teachers who are autonomously motivated are very passionate and committed to teaching and when preparing the content to be taught they consider the interests of their students, communicate their expectations clearly, and provide help and encouragement [26]. Therefore, autonomous motivation should increase engagement in structuring and autonomy-supportive teaching, whereas it would diminish controlling and chaotic teaching styles.

As for the third research hypothesis, it is only partially confirmed. In particular, self-efficacy directly predicts motivating teaching styles, suggesting that teachers who perceive themselves as capable of selecting and implementing educational and teaching strategies that respond to students’ individual needs use more structured supportive teaching strategies, consistent with previous studies. Furthermore, these results are consistent both with the Cognitive Social Theory [3] which suggests that self-efficacy can directly predict behavior, and with studies conducted in the context of SDT [7] which indicates that the need to feel effective is associated with the use of motivating styles.

Instead, although in the correlation matrix TEIP is negatively associated with demotivating styles, suggesting that as self-efficacy decreases, chaotic and control practices increase, these relationships in the full model lose their meaning. These results are similar to those found in previous studies [15,24] in which no associations were found between teacher effectiveness and controlling and chaotic teaching. However, the relationship between TEIP and (de)motivating teaching styles would seem better explained by the mediating role played by autonomous motivation (fourth research hypothesis). The results suggested that autonomous motivation may act as a buffer against the adoption of controlling and chaotic approaches, promoting the use of autonomy-supportive and structuring strategies, consistent with previous studies [14,26].

Overall, coherent with the Social Cognitive Theory [3] and SDT [7], this study suggested the role of autonomous motivation and teachers’ self-efficacy for inclusive practices as factors associated with teaching styles in inclusive education.

### 4.1. Study Limitations and Future Research

Although these findings provide a better understanding of the mediating mechanism in the relationship between TEIP and teaching styles, several limitations should be mentioned. Firstly, the research results cannot be generalized as the sample was composed only of Italian pre-service teachers. Therefore, future studies should consider a cross-cultural comparison. Secondly, the cross-sectional design of this study makes it difficult to establish cause and effect relationships. Future longitudinal or experimental studies could facilitate more causal evaluations.

Third, the unique use of self-assessment measures increases the likelihood that participants provided socially desirable responses. Therefore, future research may use other evaluation methods, such as direct observation. Fourth, other variables—such as job-related emotions, perceived social support, and job satisfaction—which may explain the TEIP–teaching styles link, should be considered. Furthermore, future research could also consider how the characteristics of pupils with special educational needs (e.g., type or severity of disability, ADHD, foreign students, etc.) affect teachers’ self-efficacy, motivation, and teaching styles.

These limitations require the findings of the present study to be interpreted carefully.

### 4.2. Practical Implication

On the basis of the aforementioned results, training programs should place more emphasis on developing teachers’ abilities in reflecting on autonomous motivation [14,24,46]. This could be a first step to encourage the adoption of more functional behaviors toward their students. Thus, support and respect for teachers’ needs for autonomy become particularly important in promoting autonomy-supportive teaching. It seems that teachers would be more prone to understanding the importance of supporting their students’ autonomy and more ready to learn how to better support autonomy if their own autonomy as teachers is reinforced as well. This suggests that in training and reform processes for developing teachers’ predispositions to support students’ autonomy, trainers, principals, and reform agents should themselves behave toward teachers in a supportive manner. For example, it is crucial that principals and reform agents provide a valid justification for teachers to engage in autonomy-supportive teaching style, leaving them the freedom to choose the most appropriate autonomy support methods in the classroom and allowing them to raise concerns and negative sentiments about the importance of autonomy-supportive teaching.

However, even if it is extremely important to a high-quality education, the link between autonomous motivation and motivating teaching styles alone is not enough; the most powerful source of supportive and inclusive education consists in the interplay between other factors, such as high self-efficacy [15,47,48,49].

## Figures and Tables

**Figure 1 ijerph-19-10921-f001:**
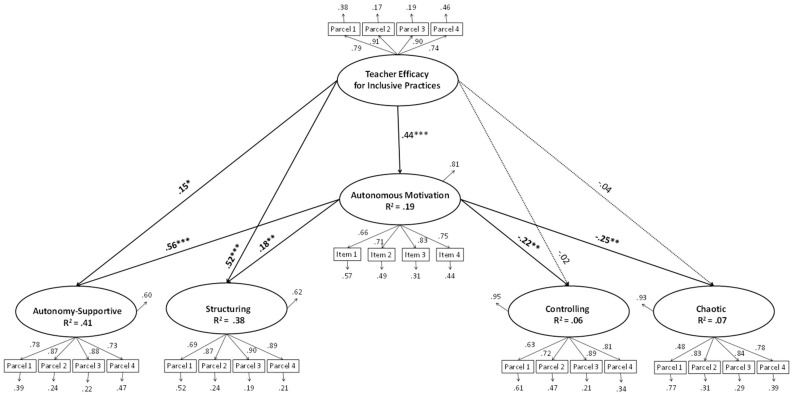
Path diagram depicting the relationships between study variables. Note: *** *p* ≤ 0.001, ** *p* ≤ 0.01, * *p* ≤ 0.05. Coefficients shown are standardized path coefficients. Dotted lines represent non-significant parameters.

**Table 1 ijerph-19-10921-t001:** Descriptive statistics and correlational analysis.

	M	DS	Skewness	Kurtosis	1	2	3	4	5	6
**Teacher Efficacy for Inclusive Practices**	4.66	0.73	−1.13	3.04	*α* = 0.94					
2. **Autonomous Motivation**	4.35	0.68	−1.49	3.13	0.42 **	*α* = 0.82				
3. **Autonomy-supportive**	5.68	0.93	−1.39	3.73	0.38 **	0.54 **	*α* = 0.91			
4. **Structuring**	5.78	0.93	−1.87	6.00	0.40 **	0.50 **	0.83 **	*α* = 0.92		
5. **Controlling**	2.46	0.95	0.95	1.33	−0.10 *	−0.16 **	−0.25 **	−0.12 *	*α* = 0.88	
6. **Chaotic**	2.04	0.80	1.80	5.95	−0.15 **	−0.19 **	−0.35 **	−0.38 **	0.59 **	*α* = 0.87

Note: *N* = 423; ** *p* ≤ 0.01, * *p* ≤ 0.05.

**Table 2 ijerph-19-10921-t002:** Path estimates, SEs and 95% CIs.

	β	SE	Lower Bound (BC) 95% CI	Upper Bound (BC) 95% CI
*Direct Effect*				
Teacher Efficacy for Inclusive Practices → Autonomous Motivation	0.44	0.07	0.26	0.54
Teacher Efficacy for Inclusive Practices → Autonomy-Supportive	0.15	0.07	0.03	0.32
Teacher Efficacy for Inclusive Practices → Structuring	0.18	0.07	0.06	0.35
Teacher Efficacy for Inclusive Practices → Controlling	−0.02	0.08	−0.18	0.15
Teacher Efficacy for Inclusive Practices → Chaotic	−0.04	0.06	−0.14	0.08
Autonomous Motivation → Autonomy-Supportive	0.56	0.09	0.53	0.91
Autonomous Motivation → Structuring	0.52	0.10	0.42	0.80
Autonomous Motivation → Controlling	−0.22	0.08	−0.43	−0.09
Autonomous Motivation → Chaotic	−0.25	0.07	−0.34	−0.08
*Indirect effect via Autonomous Motivation*				
Teacher Efficacy for Inclusive Practices → Autonomy-Supportive	0.24	0.06	0.18	0.41
Teacher Efficacy for Inclusive Practices → Structuring	0.23	0.05	0.15	0.35
Teacher Efficacy for Inclusive Practices → Controlling	−0.10	0.04	−0.19	−0.03
Teacher Efficacy for Inclusive Practices → Chaotic	−0.11	0.03	−0.15	−0.03

Note: SE = standards errors; BC 95% CI = bias corrected-confidence interval.

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
