# Peer review of "Special Education Teachers: The Role of Autonomous Motivation in the Relationship between Teachers’ Efficacy for Inclusive Practice and Teaching Styles"

_ijerph, 2022, doi:10.3390/ijerph191710921_

Round 1
Reviewer 1 Report
The content of the article is of interest and the approach is appropriate.
At time of publication, I would consider the following changes to be necessary:
1. Greater depth in the theoretical framework, adding more recent studies on the object of study.
2. A better description of the design used, its denomination and suitability.
3. More extensive discussion of the results and in accordance with the results. Use of theoretical support.
4. Revision of the form of in-text citations indicated in the journal.
Author Response
Find attached the replay to reviewer 1

Reviewer 2 Report
You can describe research objects that come from different regions, social and economic characteristics as a significant comparison

Author Response
Find attached the replay to reviewer 2

Reviewer 3 Report
The research seems up-to-date and interesting in terms of its subject. Although the research title is too long, it is suitable for the purpose.
It was also not good that the abstract of this research was not in the form of a structured summary. In other words, a summary should have been written in the abstract, including the purpose, method, data collection tool and summary of the results. However, numeric values ​​should not be included in the abstract.
The introduction of the research is appropriate in terms of literature.
The introduction part of the research is sufficient in terms of subject area. The bibliographies used are up-to-date. Therefore, the use of new bibliography in the introduction and discussion sections of the research has enriched the research.
Purpose and sub-objectives were not written in line with the findings. In addition, the importance of the research should be written before the method. The research method was not written. What kind of research method was used in the study. Care was taken to write the tables used in the research in the form of APA6 standard.
In Table 1, "Kurtosis" should be written clearly.
Third language is important in research and authors should not use the term "we".
New references from the current 2021 and 2022 years should also be used in the introduction and discussion of the research.
The discussion, conclusion, and recommendations section of the research was written as a single section. It would be better if these sections were written separately.
Author Response
Find attached the replay to reviewer 3

Reviewer 4 Report
I obviously enjoyed reading this manuscript. I believe it has merit and holds good promise. It contributes to our understanding of the issue in focus. That said, the manuscrip would benefit from revisions regarding the following aspects as I outline below:
1. There seems to be a need for more in-depth discussion on the relationship between motivation, efficacy, and style. Whilst the authors illustrated their potential relationships in the introduction, there was a lack of in-depth elaboration about what might contribute to such relationships as well as about other possibilities of the relationships between them. For instance, efficacy seems, in many cases, to be identified or considered as a mediating variable -- obviously this was not the case in the current study. What I mean here is that the authors may as well demonstrate their awareness of other possibilities of a different relationship or different relationships between the variables that were being examined. Besides, a minor issue, subsection 1.1 can not stand alone as the only subsection of secion 1.
2. Please highlight your research question(s). Have it/them more explicitly stated.
3. As to instruments, please also add more about the reasons why you have chosen the instruments that are currently used, as there are actually numerous alternatives for the three variables being examined.
4. The discussion section seems a bit weak in two ways: (1) it may be better if the authors draw more from the explanations of the data and discern more interpretations regarding, in particular, teacher learning or factors that contribute to teacher learning with relation to the three variables; (2) please provide more recommendations for future research and end your paper with such suggestions rather than the limitations. It seems always a better way to end the paper strong -- limitations are then not a good choice.
Author Response
Find attached the replay to reviewer 4

Round 2
Reviewer 2 Report
The main idea in this research can be used as a starting point for more specific further research because basically it has a very interesting object.
Reviewer 3 Report
The authors corrected the given suggestions. I congratulate for a good manuscript.